# Customizable Combination of Parameter-Efficient Modules for Multi-Task Learning

**Haowen Wang**,[*] **Tao Sun**,[*] **Congyun Jin, Yingbo Wang**
**Yibo Fan, Yunqi Xu, Yuliang Du, Cong Fan**[†]
Ant Group, Shanghai, China
{wanghaowen.whw,suntao.sun,jincongyun.jcy,wangyingbo.wyb,
fanyibo.fyb,xuyunqi.xyq,duyuliang.dyl,fancong.fan}@antgroup.com

## Abstract

Modular and composable transfer learning is an emerging direction in the field of Parameter Efficient Fine-Tuning, as it enables neural networks to better organize various aspects of knowledge, leading to improved cross-task generalization. In this paper, we introduce a novel approach Customized Polytropon (`C-Poly`) that combines task-common skills and task-specific skills, while the skill parameters being highly parameterized using low-rank techniques. Each task is associated with a customizable number of exclusive specialized skills and also benefits from skills shared with peer tasks. A skill assignment matrix is jointly learned. To evaluate our approach, we conducted extensive experiments on the Super-NaturalInstructions and the SuperGLUE benchmarks. Our findings demonstrate that `C-Poly` outperforms fully-shared, task-specific, and skill-indistinguishable baselines, significantly enhancing the sample efficiency in multi-task learning scenarios.

## 1 Introduction

As the number of parameters in Large Language Models (LLMs) continues to grow, training these models efficiently with limited computational resources has become a challenge. In recent years, there has been a shift towards employing Parameter Effective Fine-Tuning (PEFT) methods to address this issue. Examples of such methods include LoRA (Hu et al., 2022), AdaLoRA (Zhang et al., 2023a), and $(IA)^3$ (Liu et al., 2022a). These methods focus on fine-tuning the adapter while freezing the pre-trained model, effectively reducing the computational cost. By selectively updating only a portion of the model parameters, PEFT methods enable efficient training and utilization of large foundation models. This line of approaches allows for more effective use of resources while maintaining the performance of the pre-trained model on downstream tasks (Hu et al., 2022). However, despite the popularity and widely adoption of PEFT methods, the learning effectiveness of such methods, especially in multi-task scenarios, is under explored.

LLMs are famous for their extraordinary capabilities on solving multiple tasks in zero-shot or few-shot manners (Brown et al., 2020). Basic PEFT methods mentioned earlier don't take the multitask essence of real-world data into account and rely heavily on the base foundation model's capacities on the multitask generalization. Building upon the basic PEFT methods, various training approaches designed for Multi-Task Learning (MTL) have been proposed (Pfeiffer et al., 2020; Vu et al., 2021; Asai et al., 2022; Chronopoulou et al., 2023; Zadouri et al., 2023). One simple solution is to perform multitask training by training a large model on a combination of multiple tasks. This involves training the model on the union of training tasks and subsequently evaluating its performance on different testing tasks (Ye et al., 2021; Liu et al., 2022a). However, this approach overlooks the relationships between the tasks and is vulnerable to negative transfer, where the gradients associated with different tasks are misaligned (Wang et al., 2020). This misalignment of gradients can lead to sub-optimal performance and hinder the effective utilization of learned knowledge across tasks.

---

[*]Equal contribution.
[†]Corresponding author.

To enhance sample efficiency, MoLoRA (Zadouri et al., 2023) has been introduced. MoLoRA successfully applies the Mixture-of-Expert (MoE) architecture to PEFT methods and improves the model's generalization capacity across various tasks by jointly learning multiple LoRA instances. MoLoRA views each LoRA as a lightweight expert, following MoE framework, and thus allows for more specialized adaptation to different tasks from shared knowledge learnt through parallel instances.

In a recent work, Ponti et al. (2022) developed Polytropon (`Poly`) to tackle the challenges associated with multitask learning. The central idea of `Poly` is to consider each task-specific adapter as a composition of reusable skillset of basic adapters or modules. Specifically, `Poly` jointly learns an inventory of adapters (for example, LoRA) and a simple routing vector that selects and combines a variable-size subset of adapters for each individual task. This approach significantly improves the efficiency of sample sharing and utilization between multiple tasks. Although not mentioned explicitly in the original work (Ponti et al., 2022), we argue that each adapter in `Poly` can be viewed as a lightweight expert and then the whole structure follows the same pattern as Multi-gate Mixture-of-Experts (MMoE) (Ma et al., 2018), a well-known MTL framework. To be noted, in Ponti et al. (2022) also introduced a structure called MoE-LoRA, which is a simplified version of MoLoRA where the routing is controlled by a function of hidden states instead.

Based on `Poly`, a subsequent research titled Multi-Head Routing (`MHR`) (Caccia et al., 2022) was proposed. `MHR` extends the concept of parallel thinking to low-rank decomposition and introduces further improvements to the basic unit of the adapter. By leveraging low-rank decomposition, `MHR` enhances the model's ability to generalize across different tasks. The parallel improvements in the adapter's basic units allow for more efficient adaptation to different tasks while still benefiting from shared knowledge.

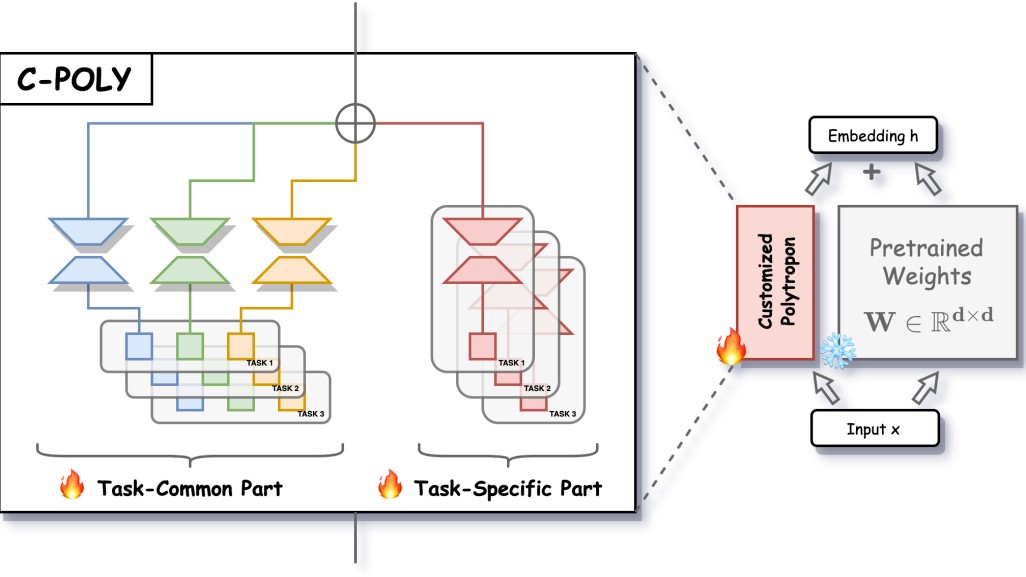

Figure 1: Overview of Customized Polytropon (`C-Poly`) framework

Inspired by Customized Sharing Control (CGC) and Progressive Layered Extraction (PLE) (Tang et al., 2020), our research has made additional improvements over `Poly`. We assume that in MTL, many tasks do share transferable knowledge, while each of them requires discriminative abilities. Based on this assumption, we explicitly divide modular skills into task-common skill modules and task-specific skill modules. We propose **Customized Polytropon (C-Poly)**, where for each task, two components are learnt jointly, a task-specific skill module and a task-common skill module. This allows each task to be characterized by not only a shared task-common module as in `Poly` and `MHR`, but also a unique subset of skills, mitigating the effect of negative transferring and leading to improved multi-task performance. Furthermore, `C-Poly` promotes interpretability by learning an explicit hierarchy of tasks based on the skills they select, which provides insights into the relation-

ship between tasks and the important skills needed for each task. Overall, our approach enhances the efficiency and effectiveness of modular skill multi-task learning, enabling better performance, interpretability, and resource utilization.

In Section 3, we conducted extensive experiments on multiple datasets and model architectures. The results of these experiments have consistently demonstrated our method `C-Poly` has surpassed the performance of the conventional PEFT methods and achieved state-of-the-art (SOTA) results in different multitasking scenarios.

## 2 METHODOLOGY

The proposed unified MTL framework `C-Poly` is shown in Figure 1, which aims to enhance sample efficiency for each task by leveraging strengths from all other tasks while keeping task-specific abilities. Suppose there are $T$ tasks and for each task, task-specific data input $x^t, t \in \{1, 2, \ldots, T\}$. The MoE-like structure consists of adapter modules (or experts), $\mathbf{\Phi} = \{\phi_1, \phi_2, \ldots, \phi_{|\mathbf{\Phi}|}\}$ and each adapter $\phi_i$ can be regarded as a function of the input data $x^t$. The major improvement of `C-Poly` is to explicitly categorize adapter modules $\mathbf{\Phi}$ into two separate parts:

- A task-common skillset: $\mathbf{\Phi}_A = \{\phi_1, \phi_2, \ldots, \phi_A\}$ having $A$ adapters.
- A task-specific skillset: $\mathbf{\Phi}_B^t = \{\phi_1^t, \phi_2^t, \ldots, \phi_B^t\}$ having $B$ adapters for each task $t$.

In total, there are $|\mathbf{\Phi}_A| + T \times |\mathbf{\Phi}_B^t| = A + T \times B$ adapters. For a simplified yet generalizable discussion, we would set $B = 1$ to keep only one task-specific adapter in the following experiments.

The combined output of the `C-Poly` adapter modules for each task input $x^t$ can be expressed in Equation 1 with $w_i$ representing the learnable weight of each adapter's output.

$$\underbrace{\sum_{i=1}^{A} w_i^t \phi_i(x^t)}_{\text{Task-Common}} + \underbrace{\sum_{j=1}^{B} w_j^t \phi_j^t(x^t)}_{\text{Task-Specific}} = \sum_{i=1}^{A} w_i^t \phi_i(x^t) + w^t \phi^t(x^t) \quad (1)$$

In the task-common part, the set of adapters $\phi_i$ are shared across all tasks, while the weights $w_i^t$ are exclusive to each task $t$. In contrast, both the weights $w_i^t$ and adapters $\phi_i^t$ are customized for each individual task $t$ in the task-specific part.

Following the notation above, various MoE-like PEFT structures can be mathematically formulated together in Table 1. Both the MoE and MMoE models only consist of the task-common part of Equation 1: in the conventional MoE approach, tasks are not differentiated, leading to shared parameters across all tasks; the MMoE framework assigns task-specific weights for each individual task, while still maintaining a shared pool of experts or adapters.

Table 1: Comparison between different MoE-like PEFT methods

| MoE Structures | PEFT Methods | Task Output |
|---|---|---|
| Conventional MoE | MoLoRA (Zadouri et al., 2023), MoE-LoRA (Ponti et al., 2022) | $\sum_{i=1}^{A} w_i \phi_i(x^t)$ |
| MMoE (Ma et al., 2018) | `Poly` (Ponti et al., 2022), `MHR` (Caccia et al., 2022) | $\sum_{i=1}^{A} w_i^t \phi_i(x^t)$ |
| CGC (Tang et al., 2020), PLE (Tang et al., 2020) | **Our Method** `C-Poly` | $\sum_{i=1}^{A} w_i^t \phi_i(x^t) + w^t \phi^t(x^t)$ |

In `C-Poly`, the weights associated with each adapter can be represented together as one allocation matrix $\mathbf{W} \in \mathbb{R}^{T \times (A+T)}$ when $B = 1$. This matrix can be further decomposed into two distinct

components: $\boldsymbol{W}_A \in \mathbb{R}^{T \times A}$ and $\boldsymbol{W}_B \in \mathbb{R}^{T \times T}$:

$$\boldsymbol{W} = [\ \boldsymbol{W}_A\ |\ \boldsymbol{W}_B\ ] \tag{2}$$

$$= \begin{bmatrix} w_1^1 & w_2^1 & \cdots & w_A^1 & w^1 & 0 & 0 & 0 \\ w_1^2 & w_2^2 & \cdots & w_A^2 & 0 & w^2 & \cdots & 0 \\ \vdots & & \ddots & \vdots & \vdots & & \ddots & \vdots \\ w_1^T & w_2^T & \cdots & w_A^T & 0 & 0 & \cdots & w^T \end{bmatrix} \tag{3}$$

To optimize the learning process in skill acquisition, we have employed different learning methods for each component of the allocation matrix. Additionally, we have incorporated low-rank approximations to further enhance the parameter efficiency.

## 2.1 TASK-COMMON SKILLS LEARNING

Task-common skills are universally applicable skills that all tasks can leverage. Previous research has focused on identifying the effectiveness of general skills modules for specific tasks by employing modular concepts at the structured input level inspired by cognitive mechanisms (Bengio, 2017; Ponti et al., 2022; Caccia et al., 2022; Zadouri et al., 2023). This concept has been translated into Softmax for cross-module or top-k selection in practical implementation.

Therefore, following Ponti et al. (2022) and Caccia et al. (2022), we utilize a task-common allocation matrix $\boldsymbol{W}_A \in \{0, 1\}^{T \times A}$ with a uniform initialization. This matrix is employed to achieve the soft partitioning of general skills. Each element $w_i^t$ in $\boldsymbol{W}_A$ is a binary value that indicates whether a particular task $t$ activates the adapter module $\phi_i$. However, since discrete binary matrices like $\boldsymbol{W}_A$ are non-differentiable, learning cannot be accomplished through gradient descent. To overcome this limitation, we adopt the Gumbel-sigmoid approach (Maddison et al., 2016; Jang et al., 2016), which allows us to obtain a set of continuously relaxed Bernoulli distributions. This approach guarantees both randomness and the ability to perform differentiable sampling:

$$\hat{w_i^t} = \sigma \left[ \log \frac{\sigma(w_i^t)\mathbf{u}}{(1 - \sigma(w_i^t))(1 - \mathbf{u})} \right], \quad \mathbf{u} \sim \mathcal{U}(0, 1) \tag{4}$$

## 2.2 TASK-SPECIFIC SKILLS LEARNING

Specialized skills refer to modular skills that acquire the distinctive attributes of each task. In complex and interconnected multitasking scenarios, a seesaw phenomenon commonly arises. In multitasking learning mode, there is often a trade-off between enhancing specific tasks' effectiveness and compromising others' effectiveness (Wang et al., 2020). To address this, we explicitly differentiate between shared and exclusive skill modules. This separation allows us to amplify the inherent characteristics of individual tasks.

We initialize the task-specific skill allocation $\boldsymbol{W}_B \in \mathbb{R}^{T \times T}$ as a unit diagonal matrix. Notably, during the actual training process, although only the diagonal of $\boldsymbol{W}_B$ are weights we concerned, entries off the diagonal are also subject to potential updates. These side-effects indicate that while solving the current task, the exclusive specialized skills of other tasks can be leveraged without influencing the parameter values of specialized skills for those tasks.

## 2.3 PARAMETER EFFICIENCY

We accomplished an efficient parameterization of skill modules by employing low-rank techniques. Every adapter in our `C-Poly` experiments is Low-Rank Adapter (LoRA) (Hu et al., 2022). LoRA is a straightforward yet effective structure specifically tailored for Transformer-based models (Vaswani et al., 2017). The idea behind LoRA is relatively straightforward. LoRA decomposes each weight matrix of the linear transformation in Transformers into the multiplication of two low-rank matrices. In other words, the linear projection $f : \mathbb{R}^d \to \mathbb{R}^d$ can be represented as follows, disregarding the bias part:

$$\boldsymbol{h}_{l+1} = \boldsymbol{h}_l\left[\boldsymbol{W}_l + \Delta \boldsymbol{W}\right] = \boldsymbol{h}_l\left[\boldsymbol{W}_l + \boldsymbol{W}_{down}\boldsymbol{W}_{up}\right] \tag{5}$$

Instead of $\Delta\boldsymbol{W} \in \mathbb{R}^{d \times d}$, two much smaller matrices $\boldsymbol{W}_{down} \in \mathbb{R}^{d \times r}$ and $\boldsymbol{W}_{up} \in \mathbb{R}^{r \times d}$, with $r \ll d$, are obtained through gradient descend optimization. Through the adoption of LoRA, updating each linear layer in the model only requires $2 \times r \times d$ parameters in the calculation, as opposed to the original $d \times d$ parameters. This results in a notable enhancement in parameter efficiency, enabling faster training even with limited computing resources.

As examined in Hu et al. (2022), LoRA can be applied to various components of Transformers, such as query, key, value, and feed-forward layers, while the choice of rank $r$ does not hold significant importance. This suggests that LoRA exhibits versatility in its applicability. In our experiments, we patched all query, key, and value layers with `C-Poly`, a combination of multiple LoRAs.

## 3 EMPIRICAL EXPERIMENTS

### 3.1 EXPERIMENTAL SETUP

In order to evaluate the efficacy of our proposed unified MTL framework `C-Poly`, which incorporates both task-common and task-specific skills, we conducted experiments on two publicly available multitasking benchmarks: SuperGLUE (Wang et al., 2019) and Super Natural-Instructions (Super NI) (Wang et al., 2022). The SuperGLUE benchmark is a widely acceptable benchmark for evaluating general-purpose language understanding. In our experiments, 7 distinct tasks were selected from the benchmark that can be effectively evaluated using the accuracy metric. The Super NI dataset, as a meta-dataset (Triantafillou et al., 2019), covers a wide range of 76 distinct task types within the field of natural language processing and comprises over 1,600 diverse NLP tasks. During the experiments, 100 tasks were randomly selected, and for each task, 1000 samples were randomly selected for training and another 100 were selected for evaluation purpose. To ensure comparability, our sampling method follows the identical approach as described in Ponti et al. (2022). To evaluate the effectiveness of the trained model, we employed various metrics for all selected tasks, including Exact Match (EM) and Rouge metrics (Lin, 2004), including Rouge-1, Rouge-L, and Rouge-LSum.

To verify the universal effectiveness of our multitasking learning approach `C-Poly`, we chose T5 Version 1.1 - LM Adapted (T5) (Raffel et al., 2020), FLAN-T5 (Chung et al., 2022) and GLM (Du et al., 2021) as the base models.

In our study, we thoroughly compared our proposed approach, `C-Poly`, and several existing MoE-like PEFT methods. The methods we compared against include LoRA, MoE-LoRA, `Poly` and `MHR`. The comparison allowed us to analyze and evaluate the performance and effectiveness of our framework relative to these established PEFT methods.

### 3.2 TRAINING DETAILS

In our experiments, we applied PEFT methods to all query, key, value matrices within every attention layer in the base models. In the case of vanilla LoRA, we set the rank of the low-rank approximation, $r = 8$. For all MoE-like tuning methods, we utilized in total 4 parallel LoRAs (experts) with $r = 2$. In `C-Poly`, we set $A = 3$ LoRA for task-common skills and $B = 1$ LoRA for task-specific skills. This decision was made to ensure a comparable number of training parameters across all methods.

We trained our model with cross entropy loss for only 1 epoch, and set batch size of 4 on both Super NI and SuperGLUE datasets during training. The AdamW optimizer (Loshchilov & Hutter, 2017) was used, with a learning rate of $5e^{-5}$. We also employed the linear decay strategy (Loshchilov & Hutter, 2016) as the learning rate scheduler with a weight decay of 0.01 and a warmup ratio of 0.06. All experiments were conducted on a single NVIDIA Tesla A100 graphics card.

### 3.3 MAIN RESULTS AND DISCUSSION

#### 3.3.1 ANALYSIS OF BALANCED MULTITASK LEARNING

In Figure 2, we present a comparative analysis of various fine-tuning methods across multiple tasks within the SuperGLUE benchmark. When tuning with full parameters (FT), the overall average accuracy is the lowest among all approaches because of relatively poor performance in the MultiRC sub-task. This phenomenon, known as the seesaw effect, a manifestation of the negative transfer

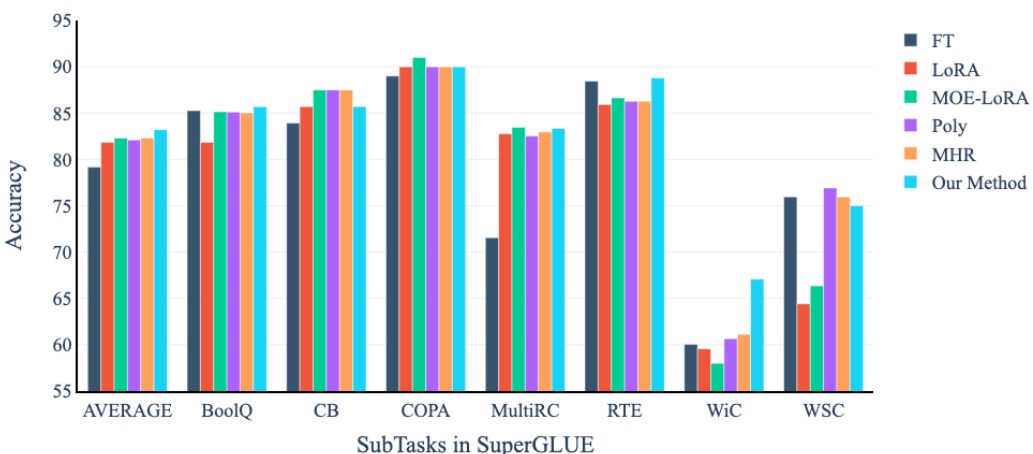

Figure 2: FLAN-T5-Large with different PEFT methods on SuperGLUE benchmark, compared with Full Fine-tuning (FT). We reported overall averaged (AVERAGE) and task-specific accuracy for all sub-tasks.

problem when tackling multiple tasks concurrently, holds significant importance in the domain of multi-task learning (Pan & Yang, 2010; Sun et al., 2017; Tang et al., 2020). Our method `C-Poly`, on the other hand, demonstrates constant improvement over all sub-tasks thanks to the task-specific skill learning module. The results reveal that `C-Poly` can effectively mitigate the negative transfer and seesaw effect issues. As shown in Appendix A.4, we conducted experiments on FLAN-T5-XL (2B) and found that the base model with larger amount of parameters has a stronger fitting ability for multiple tasks. Consequently, the experimental results showed that our method can bring stable improvements in reducing negative migration.

Table 2: FLAN-T5-Large and GLM-10B with different adaptation methods on the SuperGLUE benchmark. We report the overall (matched and mismatched) accuracy for BoolQ, CB, COPA, MultiRC, RTE, WiC and WSC. Higher is better for all metrics.

| Base Model | PEFT Method | AVG | BoolQ | CB | COPA | MultiRC | RTE | WiC | WSC |
|---|---|---|---|---|---|---|---|---|---|
| FLAN-T5-Large | LoRA | 81.85 | 81.85 | 85.71 | 90.00 | 82.78 | 85.92 | 59.56 | 64.42 |
| | MOE-LoRA | 82.31 | 85.14 | **87.50** | **91.00** | **83.46** | 86.64 | 57.99 | 66.35 |
| | Poly | 82.09 | 85.11 | **87.50** | 90.00 | 82.53 | 86.28 | 60.66 | **76.92** |
| | MHR | 82.31 | 85.02 | **87.50** | 90.00 | 82.96 | 86.28 | 61.13 | 75.96 |
| | **Our Method** | **83.21** | **85.69** | 85.71 | 90.00 | 83.35 | **88.81** | **67.08** | 75.00 |
| GLM-10B | LoRA | 52.05 | 60.98 | 46.38 | 65.70 | 62.43 | 57.37 | 39.15 | 32.32 |
| | MoE-LoRA | 53.86 | 63.31 | 45.02 | 63.41 | 64.01 | 61.22 | 40.35 | 39.66 |
| | Poly | 56.99 | 64.65 | 52.17 | 65.54 | 65.66 | 62.15 | 41.71 | 47.08 |
| | MHR | 56.92 | 64.85 | 50.79 | 66.36 | 65.79 | 62.75 | 42.35 | 45.58 |
| | **Our Method** | **62.26** | **67.31** | **60.38** | **70.04** | **67.90** | **68.01** | **48.71** | **53.42** |

### 3.3.2 EVALUATION ON MORE MODELS AND MORE TASKS

The effectiveness of different PEFT methods are evaluated on two architectures: T5 and GLM. Table 2 presents the performance of these methods on a dataset consisting of 7 tasks from the SuperGLUE. We display the evaluation results for the 7 sub-tasks individually, as well as their average performance. Table 3 compares the performance of different PEFT methods on the SuperNI. The indicators in the table represent the average performance evaluation over 100 tasks.

Table 3: FLAN-T5-Large, T5-Large and GLM-10B with different adaptation methods on the 100 randomly selected tasks from SuperNI dataset. We report the average Rouge-1, Rouge-L, and Rouge-LSum for all tasks. Higher is better for all metrics.

| Base Model | PEFT Method | Rouge-1 | Rouge-L | Rouge-LSum |
|---|---|---|---|---|
| | LoRA | 68.26 | 67.42 | 67.42 |
| | MoE-LoRA | 68.59 | 67.76 | 67.75 |
| FLAN-T5-Large | Poly | 68.45 | 67.60 | 67.58 |
| | MHR | 68.84 | 67.77 | 67.78 |
| | **Our Method** | **68.69** | **67.80** | **67.82** |
| | LoRA | 34.16 | 33.64 | 33.65 |
| | MoE-LoRA | 36.82 | 36.13 | 36.15 |
| T5-Large | Poly | 43.04 | 42.05 | 42.09 |
| | MHR | 44.24 | 43.32 | 43.34 |
| | **Our Method** | **49.34** | **48.50** | **48.51** |
| | LoRA | 43.16 | 42.04 | 42.09 |
| | MoE-LoRA | 45.97 | 44.79 | 44.89 |
| GLM-10B | Poly | 47.96 | 46.80 | 46.80 |
| | MHR | 48.53 | 47.34 | 47.33 |
| | **Our Method** | **49.53** | **48.45** | **48.45** |

The results are two-folded. Firstly, it highlights that MoE-LoRA consistently demonstrates improvement over LoRA, attributed to the enhanced parameter flexibility from the MoE structure. We compared FLAN-T5-Large, T5-Large, and GLM-10B, and the results showed that it can significantly alleviate the phenomenon of negative migration in both architectures and improve the flexibility of adaptive parameters compared to `LoRA` and `Poly`. In the comparison of FLAN-T5-Large, although it has undergone a large number of pre-training with similar instruction samples, our method can still bring some improvement. Moreover, in Appendix A.2 we conducted the ablation experiments on a broader range of models including T5-Large (0.78B), T5-XL (3B), and stronger one T5-XXL (11B), as well as their FLAN-T5 counterparts, and the results showed that our methods remained robust and significant as model parameters increased.

Secondly, the results demonstrate that our method `C-Poly` has achieved optimal performance in both architectures and on the SuperGLUE and SuperNI datasets. The ablation experiments in Appendix A.1 on the number of tasks (10-50-100) conducted on GLM-10B and FLAN-T5-Large indicate that our method is still significantly effective when the number of tasks increases. The explicit separation of task-specific skills and task-common skills in our design enables the skill modules to capture task-specific differences while sharing abstracted general skill modules effectively. Due to the explicit separation, the negative transfer phenomenon has been significantly reduced as in Tang et al. (2020), which can be verified on both datasets and becomes more pronounced as the number of learning tasks increases.

### 3.3.3 PARAMETER EFFICIENCY ANALYSIS

In Figure 3, we assessed PEFT methodologies across varying parameter magnitudes, gauging their efficacy on the SuperGLUE benchmark with FLAN-T5-Large as the base model. Furthermore, we compared their performance against that derived from full parameter fine-tuning. It is evident that our approach attains better performance among all PEFT methodologies with identical scales of parameters and training epochs. Notably, `C-Poly` even outperforms other methods with more parameters. Our methods explicitly segregate shared and proprietary skills and effectively ensure

parameter efficiency in multi-task learning under equivalent training settings. This trend is also observed in the SuperNI dataset evaluation. As a result, we argue that adopting both task-common skills learning and task-specific skills learning as a paradigm constitutes a robust strategy for achieving parameter efficiency in adapting multiple tasks.

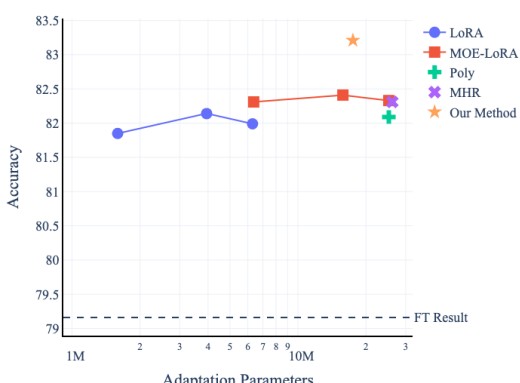

Figure 3: Accuracy of PEFT methods and Full Fine-tuning on SuperGLUE benchmark when using T5-Large as the base model. The X-axis shows the number of trainable parameters.

### 3.3.4 DEEPER INSIGHTS INTO `C-Poly`

As outlined in Section 3.3.2, our dual-skill framework, `C-Poly`, consistently delivers robust enhancements across various architectures, task scales, and model sizes. Here, we try to explore the intrinsic attributes of `C-Poly` that drive these performance gains.

**Explicit skill separation enhances knowledge sharing** Task-specific skills enable the task-common component to focus on task similarities, while they manage the distinct elements of each task. In Appendix A.5, we conducted a deep analysis on the task-common allocation matrix $W_A$. Figures 5, 6, and 7 demonstrate $W_A$ of `C-Poly` in certain layers of GLM-10B for SuperNI-10, SuperNI-50, and SuperNI-100 respectively. After proper normalization, these matrices show clear differences in skill allocation for different tasks. Additionally, we performed task clustering based on the skill allocations learned in all layers of the GLM-10B model trained on SuperNI-100, with and without the task-specific component (i.e., comparing `C-Poly` and `Poly`). The clustering outcomes are shown as dendrograms in Figures 8 and 9, which suggest the enhanced performance of `C-Poly` is due to a more balanced task hierarchy, facilitating more effective knowledge transfer among similar tasks and improved distinction of unrelated ones.

**Equilibrium in skill allocation is crucial** The introduction of `C-Poly` prompts the need to balance task-common and task-specific parameter allocation. We must establish the optimal $A$ and $B$ values. In Appendix A.3, we explore various $(A, B)$ pairings, maintaining a fixed total parameter count and confirm that task-specific skills significantly boost model performance. However, allocating enough parameters to task-common skills is crucial for shared knowledge acquisition. Excessive focus on task-specific parameters can impede the learning process, potentially causing overfitting and hindering the model's ability to recognize similarities across tasks, which may lead to suboptimal performance. A balanced parameter distribution promotes clear task distinction and helps prevent overfitting, preserving the model's generalization capabilities. Identifying the ideal parameter ratio and configuration is a critical aspect for `C-Poly` and may vary depending on the nature of the tasks and datasets involved.

## 4 RELATED WORKS

### 4.1 PARAMETER-EFFICIENT FINE-TUNING

Numerous researchers have proposed incorporating adapters within neural networks, strategically placed between existing layers, and enforcing weight constraints on these adapters. LoRA (Hu et al., 2022), for instance, advocates for fine-tuning the model by learning low-rank matrix weights and aligning them with sovereignty. Building upon LoRA, (IA)$^3$ (Liu et al., 2022a) offers further enhancements by introducing a relatively modest number of novel parameters. In the context of model adjustment, Prefix Tuning (Li & Liang, 2021), along with Prompt Tuning (Lester et al., 2021) and P-Tuning (Liu et al., 2022b), emerges as a technique that exclusively optimizes a small segment of continuous task-specific vectors, thereby strengthening downstream task optimization.

### 4.2 MODULAR MULTI-TASK LEARNING

Modular neural network architectures (Jacobs et al., 1991a) offer benefits like positive transfer, compositionality, and parameter efficiency (Zhang et al., 2023b). They consist of modules (which represent skills that can be combined and updated independently), a routing function (to select modules per task or example with variants like fixed, learned hard, or learned soft routing (Rosenbaum et al., 2017; Jacobs et al., 1991b; Fernando et al., 2017)), and an aggregation function (to merge outputs of active modules, often as a learnable network). Notably, within the realm of NLP, Fedus et al. (2021)successfully extended the pre-training of large language models to trillions of parameters by leveraging the MoE architecture. Previous studies (Rajendran et al., 2015; Ponti et al., 2022; Kingetsu et al., 2021; Kudugunta et al., 2021) have explored approaches to enforce parameter reuse and modularity in multitasking learning. Rajendran et al. (2015) trained individual modules for each task and subsequently learned how to reuse these modules.

### 4.3 LANGUAGE MODELS

The Transformer architecture serves as a fundamental framework for sequence pair modeling. Building upon this foundation, Radford & Narasimhan (2018) employed a stack of Transformers to effectively model autoregressive languages through the deployment of encoders and decoders. BERT (Devlin et al., 2019) and GPT-2 (Radford et al., 2019) are classic text modeling methodologies, both relying on Transformer units pre-trained on vast amounts of textual data. The encoder-only architecture model is particularly suited for comprehension-based tasks, whereas generative tasks benefit from both encoder-decoder and decoder-only architecture models due to their autoregressive nature (Fu et al., 2023; Sarrouti et al., 2022).

## 5 CONCLUSION

In this article, we introduce a novel paradigm for PEFT called Customized Polytropon `C-Poly`. By explicitly distinguishing between task-common and task-specific skills, our method enables efficient multi-task fine-tuning on large language models, even with limited computational resources. Our approach addresses the challenge of resource limitations and allows for efficient training. The separation of exclusive and general skills effectively mitigates the seesaw problem and negative transfer commonly encountered in multitasking learning, leading to superior overall performance, which also offers compelling interpretability. Evaluations on various benchmark demonstrate the effectiveness of the proposed method, which surpasses existing PEFT baselines and achieves state-of-the-art performance. These results highlight the potential and significance of our unified multi-task learning framework `C-Poly` in the field of parameter-efficient multi-task learning.

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

# A    ADDITIONAL RESULTS

## A.1    ABLATION ON THE NUMBER OF TASKS

To understand how our approach scales with the number of tasks, we performed experiments with varying task numbers - 10, 50, and 100 from SuperNI dataset. Table 4, Table 5 and Table 6 show the performance of T5-Large, Flan-T5-Large, and GLM-10B models respectively. These results affirm that our method `C-Poly` is highly effective and scalable across various task numbers. Our approach shows not only superior performance in lower task settings but also maintains its competitive edge as the number of tasks increases. It is worth mentioning that FLAN-T5-Large is trained on FLAN dataset, and it may have some similarities or overlaps with the SuperNI dataset, which makes its performance relatively high.

Table 4: Performance of T5-Large on different numbers of tasks from SuperNI dataset. We report the average Rouge-1, Rouge-L, and Rouge-LSum for all tasks. Higher is better for all metrics.

| # of Tasks | PEFT Method | Rouge-1 | Rouge-L | Rouge-LSum |
|---|---|---|---|---|
| 10 | LoRA | 14.22 | 14.12 | 14.19 |
| | MoE-LoRA | 16.75 | 16.71 | 16.76 |
| | Poly | 17.34 | 17.31 | 17.38 |
| | MHR | 17.17 | 17.11 | 17.16 |
| | **Our Method** | **42.62** | **42.47** | **42.60** |
| 50 | LoRA | 32.58 | 31.64 | 31.59 |
| | MoE-LoRA | 39.50 | 38.47 | 38.46 |
| | Poly | 46.13 | 44.25 | 44.28 |
| | MHR | 47.27 | 45.43 | 45.39 |
| | **Our Method** | **53.39** | **51.68** | **51.63** |
| 100 | LoRA | 34.16 | 33.64 | 33.65 |
| | MoE-LoRA | 36.82 | 36.13 | 36.15 |
| | Poly | 43.04 | 42.05 | 42.09 |
| | MHR | 44.24 | 43.32 | 43.34 |
| | **Our Method** | **49.34** | **48.50** | **48.51** |

Table 5: Performance of FLAN-T5-Large on different numbers of tasks from SuperNI dataset. We report the average Rouge-1, Rouge-L, and Rouge-LSum for all tasks. Higher is better for all metrics.

| # of Tasks | PEFT Method | Rouge-1 | Rouge-L | Rouge-LSum |
|---|---|---|---|---|
| 10 | LoRA | 67.82 | 67.01 | 67.03 |
| | MoE-LoRA | 67.95 | 67.12 | 67.15 |
| | Poly | 68.10 | 67.29 | 67.33 |
| | MHR | 77.49 | 77.25 | 77.36 |
| | **Our Method** | **77.73** | **77.58** | **77.61** |
| 50 | LoRA | 70.66 | 69.10 | 69.03 |
| | MoE-LoRA | 70.81 | 69.25 | 69.21 |
| | Poly | 70.76 | 69.23 | 69.15 |
| | MHR | 70.92 | 69.39 | 69.33 |
| | **Our Method** | **71.17** | **69.68** | **69.62** |
| 100 | LoRA | 68.26 | 67.42 | 67.42 |
| | MoE-LoRA | 68.59 | 67.76 | 67.75 |
| | Poly | 68.45 | 67.60 | 67.58 |
| | MHR | 68.84 | 67.77 | 67.78 |
| | **Our Method** | **68.69** | **67.80** | **67.82** |

Table 6: Performance of GLM-10B on different numbers of tasks from SuperNI dataset. We report the average Rouge-1, Rouge-L, and Rouge-LSum for all tasks. Higher is better for all metrics

| # of Tasks | PEFT Method | Rouge-1 | Rouge-L | Rouge-LSum |
|---|---|---|---|---|
| 10 | LoRA | 30.64 | 30.40 | 30.42 |
| | MoE-LoRA | 33.92 | 33.79 | 33.77 |
| | Poly | 34.53 | 34.41 | 34.31 |
| | MHR | 33.63 | 33.47 | 33.47 |
| | **Our Method** | **43.74** | **43.72** | **43.65** |
| 50 | LoRA | 34.16 | 33.00 | 32.98 |
| | MoE-LoRA | 39.87 | 38.63 | 38.55 |
| | Poly | 44.81 | 43.09 | 43.07 |
| | MHR | 45.32 | 43.62 | 43.56 |
| | **Our Method** | **53.17** | **51.27** | **51.32** |
| 100 | LoRA | 43.16 | 42.04 | 42.09 |
| | MoE-LoRA | 45.97 | 44.79 | 44.89 |
| | Poly | 47.96 | 46.80 | 46.80 |
| | MHR | 48.53 | 47.34 | 47.33 |
| | **Our Method** | **49.53** | **48.45** | **48.45** |

## A.2 ABLATION ON THE MODEL SCALE

We conducted an ablation study across various scales of the T5 model, including T5-Large (0.78B), T5-XL (3B), and T5-XXL (11B), as well as FLAN-T5 variants. Table 7 and Table 8 summarised the experiments conducted on 100 tasks from SuperNI. The results demonstrate a clear trend: as the model scale increases, there is a consistent improvement in performance for all methods, including ours. Notably, our method `C-Poly` shows significant gains over other PEFT methods across all scales of the T5 and FLAN-T5 models, suggesting that our approach effectively leverages the increased model capacity. The performance improvement is more pronounced in larger models (T5-XXL and FLAN-T5-XXL), indicating that while model capacity plays a role, our method might be particularly effective at utilizing the additional capacity.

Table 7: Performance of T5 models on the 100 randomly selected tasks from SuperNI dataset. We report the average Rouge-1, Rouge-L, and Rouge-LSum for all tasks. Higher is better for all metrics.

| Base Model | PEFT Method | Rouge-1 | Rouge-L | Rouge-LSum |
|---|---|---|---|---|
| T5-Large | LoRA | 34.16 | 33.64 | 33.65 |
| | MoE-LoRA | 36.82 | 36.13 | 36.15 |
| | Poly | 43.04 | 42.05 | 42.09 |
| | MHR | 44.24 | 43.32 | 43.34 |
| | **Our Method** | **49.34** | **48.50** | **48.51** |
| T5-XL | LoRA | 34.93 | 34.34 | 34.40 |
| | MoE-LoRA | 39.78 | 38.83 | 38.87 |
| | Poly | 43.61 | 42.61 | 42.62 |
| | MHR | 45.53 | 44.62 | 44.61 |
| | **Our Method** | **50.57** | **49.74** | **49.76** |
| T5-XXL | LoRA | 49.97 | 48.89 | 48.93 |
| | MoE-LoRA | 52.14 | 51.12 | 51.15 |
| | Poly | 55.42 | 54.65 | 54.64 |
| | MHR | 55.81 | 55.01 | 55.01 |
| | **Our Method** | **62.23** | **61.44** | **61.47** |

Table 8: Performance of FLAN-T5 models on the 100 randomly selected tasks from SuperNI dataset. We report the average Rouge-1, Rouge-L, and Rouge-LSum for all tasks. Higher is better for all metrics

| Base Model | PEFT Method | Rouge-1 | Rouge-L | Rouge-LSum |
|---|---|---|---|---|
| FLAN-T5-Large | LoRA | 68.26 | 67.42 | 67.42 |
| | MoE-LoRA | 68.59 | 67.76 | 67.75 |
| | Poly | 68.45 | 67.60 | 67.58 |
| | MHR | 68.84 | 67.77 | 67.78 |
| | **Our Method** | **68.69** | **67.80** | **67.82** |
| FLAN-T5-XL | LoRA | 71.01 | 70.21 | 70.24 |
| | MoE-LoRA | 71.08 | 70.29 | 70.33 |
| | Poly | 71.12 | 70.31 | 70.35 |
| | MHR | 71.18 | 70.36 | 70.40 |
| | **Our Method** | **71.57** | **70.72** | **70.74** |
| FLAN-T5-XXL | LoRA | 71.89 | 71.07 | 71.08 |
| | MoE-LoRA | 72.05 | 71.25 | 71.26 |
| | Poly | 72.55 | 71.76 | 71.78 |
| | MHR | 72.40 | 71.61 | 71.63 |
| | **Our Method** | **73.09** | **72.27** | **72.28** |

## A.3 ABLATION ON THE SKILL ALLOCATION

We designed an experiment to investigate the impact of the relative size of common skill and task specific skill on the results under the same parameter quantity, as shown in Table 9. The result shows that the gain is most significant when the number of task specific skills is 1, indicating that explicitly separating the general skills and proprietary skills of skills is the main factor that brings improvement, which is consistent with the methods and viewpoints explained in our paper.

Table 9: Performance of T5-Large, FLAN-T5-Large and GLM-10B on the 100 randomly selected tasks from SuperNI dataset. For each model, we apply `C-Poly` with different settings of skill allocation. We report the average Rouge-1, Rouge-L, and Rouge-LSum for all tasks. Higher is better for all metrics.

| Base Model | Common Skills | Specific Skills | Rouge-1 | Rouge-L | Rouge-LSum |
|---|---|---|---|---|---|
| T5-Large | 4 | 0 | 43.04 | 42.05 | 42.09 |
| | 3 | 1 | **49.34** | **48.50** | **48.51** |
| | 2 | 2 | 48.79 | 47.94 | 47.94 |
| | 1 | 3 | 47.46 | 46.60 | 46.59 |
| FLAN-T5-Large | 4 | 0 | 68.45 | 67.60 | 67.58 |
| | 3 | 1 | **68.69** | **67.80** | **67.82** |
| | 2 | 2 | 68.47 | 67.63 | 67.65 |
| | 1 | 3 | 68.21 | 67.36 | 67.41 |
| GLM-10B | 4 | 0 | 47.96 | 46.80 | 46.80 |
| | 3 | 1 | **49.53** | **48.45** | **48.45** |
| | 2 | 2 | 49.19 | 48.16 | 48.16 |
| | 1 | 3 | 46.85 | 45.74 | 45.73 |

## A.4 EXPERIMENTS OF FLAN-T5-XL ON SUPERGLUE

We conducted SuperGLUE experiment on FLAN-T5-XL and the result is summarized in Figure 4. Experiments show that compared to FLAN-T5-Large (0.78B) in Figure 2, FLAN-T5-XL (2B) can achieve a more stable improvement on the SuperGLUE benchmark. As the model gets larger, its generalization ability becomes stronger and it can adapt to more tasks, meanwhile because of `C-Poly`, the phenomenon of negative transfer gets weakened.

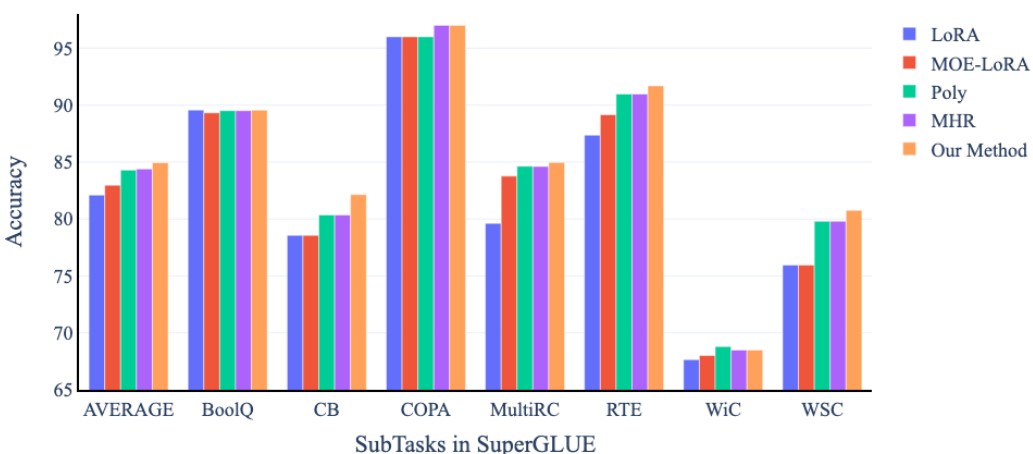

Figure 4: FLAN-T5-XL with different PEFT methods on SuperGLUE benchmark. We reported overall averaged (AVERAGE) and task-specific accuracy for all sub-tasks.

## A.5 ANALYSIS ON THE TASK-COMMON SKILL ALLOCATION

To further analyze our approach, we visualize the allocation weights for task-common skills, in Figure 5, Figure 6, and Figure 7. We reuse the GLM-10B models trained on different number of SuperNI tasks in Table 6, and select layer 0-10-20-30-40 for visualization. These figures show that the learned task-common weights for different tasks are clearly separated and differentiated.

To compare `C-Poly` with original `Poly`, we clustered tasks based on the task-common skill allocations learned in all layers of the GLM-10B model trained on SuperNI-100. The clustering results are showed in Figure 8 and Figure 9. From the dendrograms, we can see that the `C-Poly` clusters are more reasonable and balanced than those from `Poly`. It shows that our method `C-Poly` has stronger discrimination on the differences and similarities among tasks.

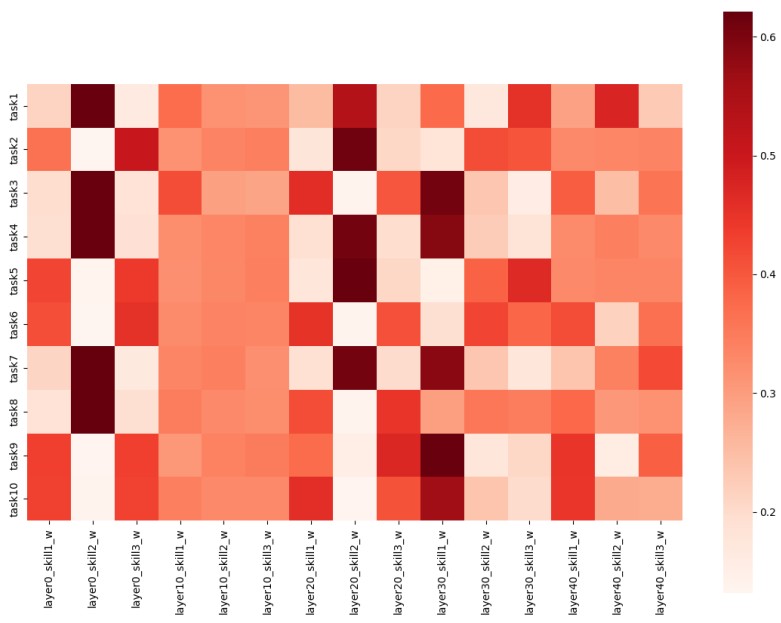

Figure 5: Visualization of the common skill allocation matrix $W_A$ of selected transformer layers in GLM-10B after training on SuperNI-10.

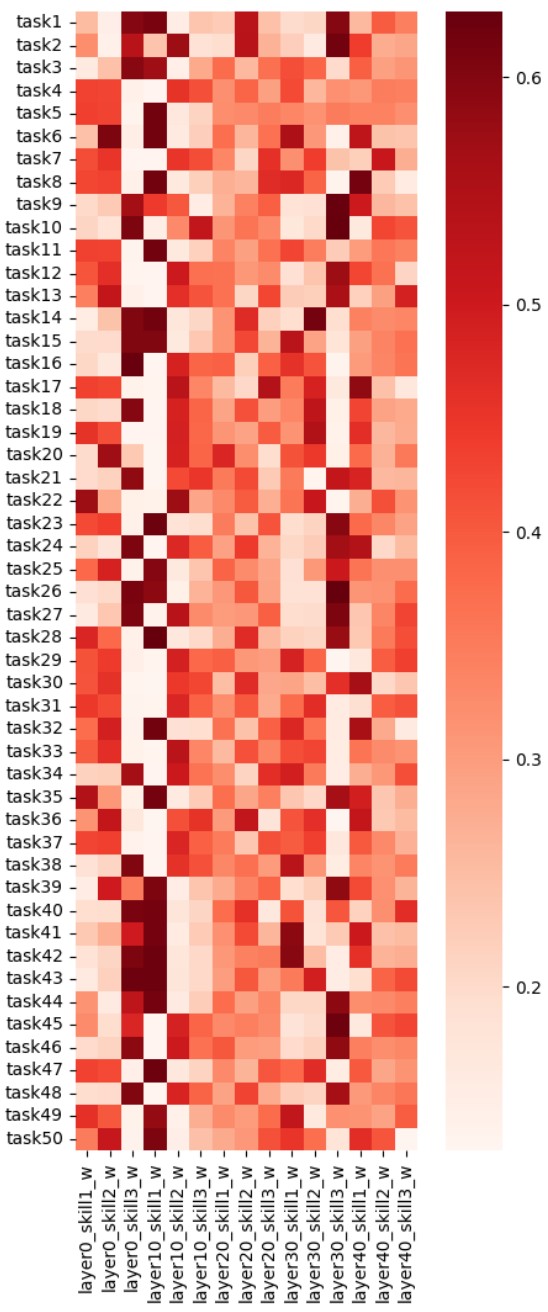

Figure 6: Visualization of the common skill allocation matrix $W_A$ of selected transformer layers in GLM-10B after training on SuperNI-50.

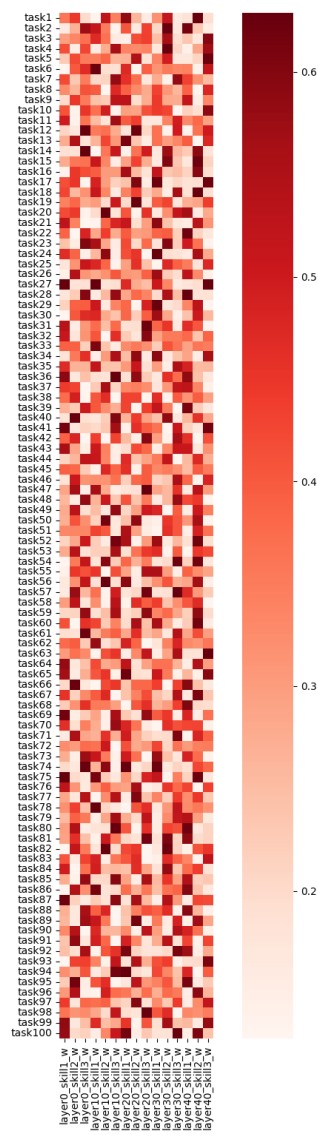

Figure 7: Visualization of the common skill allocation matrix $W_A$ of selected transformer layers in GLM-10B after training on SuperNI-100.

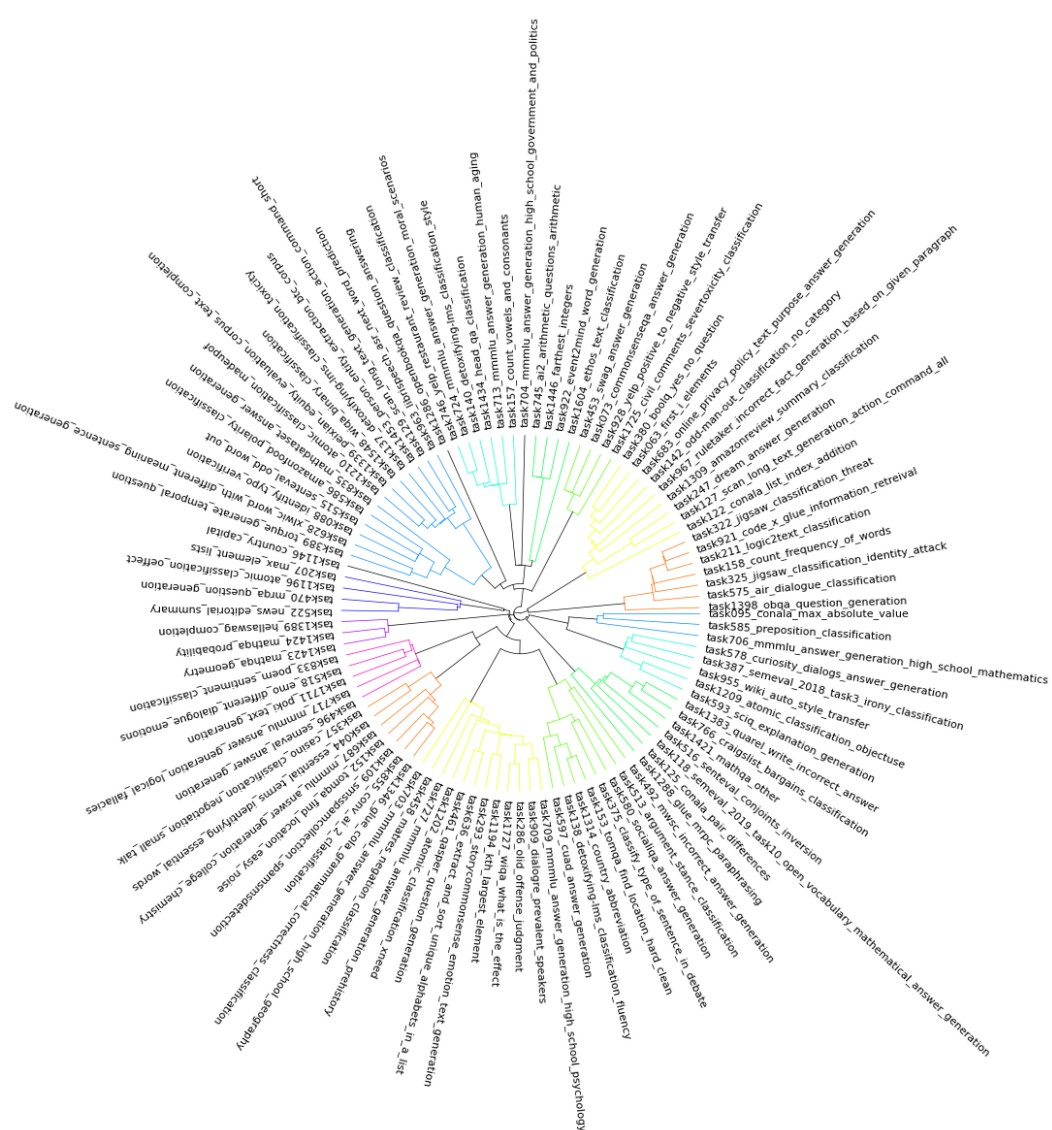

Figure 8: Task clustering dendrogram for common skill allocation matrix $W_A$ of C-Poly using GLM-10B as the base model in SuperNI-100 experiment. Tasks are grouped into the same category if they share a similar subset of skills.

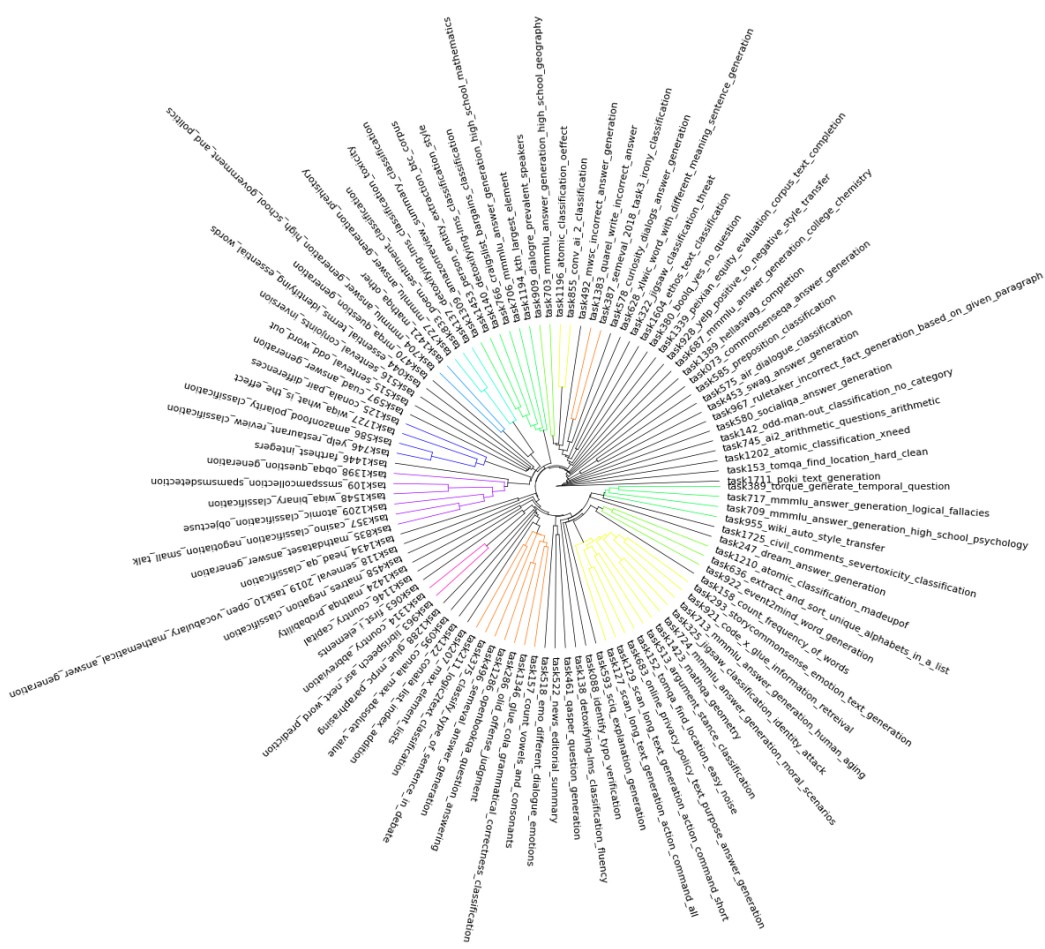

Figure 9: Task clustering dendrogram for common skill allocation matrix $W_A$ of `Poly` using GLM-10B as the base model in SuperNI-100 experiment. Tasks are grouped into the same category if they share a similar subset of skills.

