# OpenReview forum: "Customizable Combination of Parameter-Efficient Modules for Multi-Task Learning"
_ICLR.cc/2024/Conference — ICLR 2024 poster_

### Official Review · Reviewer_ZWwD · 2023-10-31

**Soundness:** 3 good
**Presentation:** 3 good
**Contribution:** 2 fair
**Rating:** 6
**Confidence:** 3

**Summary:**

The paper mixes different ideas from the multi-task learning and parameter-efficient mixture of experts literature: Methods such as `MoLora` introduced the idea of using low-rank LoRA modules as "experts". Methods such as `Poly` or `MHR` show that a new task can be learned as a combination of adapters modules (~ mixture of experts) where the combination weights are task-specific.

This paper additional proposes to explicitly capture the fact that each task can have shared knowledge (shared across all tasks) as well as task-specific knowledge at the adapters level. In this setting, a new task is now learned as a soft combination of **(i)** $A$ LoRa modules shared across all tasks and **(ii)** $B$ task-specific LoRA modules ($B$ separate modules for each task). In this case, all combination weights are task-specific, and when $B = 0$, we recover previous approaches such as `Poly`.

In the proposed framework, every adapter is a LoRA module and the combination weights are binary, i.e. each expert can be activated or not. During training, the binary decisions are approximated with Gumbel-sigmoids to allow for backpropagation.

The proposed approach is evaluated on `T5-Large` and `GLM-10B` as backbone, on the SuperGlue and Super Natural Instructions datasets, and compared to a set of recent baselines (LoRa, MoE-LoRA, Poly and MHR)

**Strengths:**

- **interesting research direction:** I think the overall goal of the paper is clear and interesting. Parameter-efficient Mixture-of-experts with LoRA modules is a recent and interesting research direction, and generalizing this framework to multi-task/domain learning as was done for other "adapters" type of work seems very natural. Similarly, the idea of separating task-specific and shared knowledge has shown some success in multi-task learning literature, and makes a lot of sense in that setup.

- **Diversity of the experimental setup:** The experiments consider two backbones, as well as two multi-task benchmarks with quite different number of tasks. The proposed method is also compared to several recent baselines

- **Experimental improvement**: While the improvement in experiments is not always consistent, it does show significant gains in some benchmarks, in particular with the decoder-only `GLM` backbone.

**Weaknesses:**

- **Explicit vs Implicit separation of task-specific and shared modules**: To my understanding,  the proposed method can be seen as subcase of some of the baselines mentioned : If the number of adapters in `Poly` is set to $A + BT$, then in theory the model could learn to make certain modules task-specific by setting $w_i^t$ to 1 for only certain pairs of $(i, t)$. In contrast, the proposed method makes this structure explicit by defining task-specific modules; However, this still requires additional $BT$ task-specific adapter modules, hence an increased number of parameters. From the experiments, it is not clear to me if the number of parameters/modules used by the baselines and the proposed method are comparable.

- **Experimental analysis**: In general, I find the proposed idea interesting, but I lack some experimental insights to better understand the proposed method. For instance, some points I found unclear are the following:
  -  1) For instance, the number of task-specific modules $B$ for each task seems to be an important parameter, but I did not see a lot of discussion/ablation about it; It seems that $B$ is set to 1 throughout the paper
  - 2) Similarly, the proposed method can use $A + B$ modules per task, while baselines only consider $A$ task-shared modules: It is not clear to me if the setup is fair to the baselines. Considering other  configurations would be interesting.
  - 3) From the results in **Table 2**, it seems that the performance improvement is not consistent: For instance with the encoder-decoder `T5` model, improvement mainly comes from the `WiC` task, which could mean this task is more likely to interfere with others. However for the decoder-only `GLM`, we see that adding a task-specific module yield significant improvement across all tasks. It is not clear to me why that is the case: Since the tasks are the same in both case, it does not seem to be directly related to the multi-task setup; could it be related to the number of LoRA modules with respect to the architecture (*see point 2*) ?

**Note post-rebuttal:** I still have some reserves about the method's robustness and sensitivity to the task specific/shared setup ratio, but in light of the rebuttal addressing my main concerns and added ablation experiments, I'll raise my score from 5 to 6

**Questions:**

- **Minor suggestion on writing:**
  * I found the introduction hard to read as it mixes introduction with some related work towards the end, and introduces several concepts without really contextualizing them (e.g. in two paragraphs, the text jumps from MoLora $\rightarrow$ Poly $\rightarrow$ MHR $\rightarrow$ CGC and PLE).
  * Using $A$ and $B$ in Equation 5 is a bit confusing as these were introduced earlier as the number of shared and task-specific modules

---

> ### Author Response · Authors · 2023-11-16
> **W1/W2.1/W2.2：Explicit vs Implicit separation of task-specific and shared modules**
>
> Thank you for your interest in our work and for pointing out areas where further experimental insights are needed. We appreciate your observations and would like to address them as follows:
>
> 1. **Number of Task-Specific Modules (B)**
>
> Our initial choice of B=1 for task-specific modules was made to streamline the model's configuration and facilitate analysis. However, in response to your suggestion, we revisited this aspect in Ablation Study 1. We explored and demonstrated the performance variations with different allocations of task-specific modules, providing insights into the impact of varying B on our model's effectiveness.
>
> 2. **Fairness in Experimental Setup**
>
> We understand the importance of ensuring a fair comparison with baselines, particularly regarding our use of A+B modules versus the A task-shared modules in baseline methods. Initially, our goal was to highlight the significance of adding task-specific modules. Furthermore, we presented an analysis of the number of trainable parameters in Figure 3 to elucidate this aspect. To foster a more equitable comparison, as mentioned in our general comment, we modified our experimental setup to (A−B + B) modules. This adjustment aligns the number of modules with baseline configurations, ensuring a more appropriate and rigorous comparison.

---

> > ### Comment · Reviewer_ZWwD · 2023-11-21
> > **Thanks for your response**
> >
> > Dear authors, thanks for your responses!
> > - The new tables solve my concern about fairness wrt to baselines and the $A$ and $B$ hyperparameters
> > - The added results varying $(A, B)$ configurations are also very interesting though they do raise the important question of tuning the ratio between these two parameters as it clearly has an impact on performance
> >
> > I still have some reserves about the method's robustness and sensitivity to the task specific/shared setup ratio, but in light of the rebuttal addressing my main concerns and added ablation experiments, I'll raise my score to 6

---

> > > ### Author Response · Authors · 2023-11-22
> > > **Thank you for your response!**
> > >
> > > Dear Reviewer,
> > >
> > > Thank you once again for your valuable time and insights. We are pleased that our additional ablation studies have effectively addressed your main concerns.
> > >
> > > We acknowledge and appreciate your reservations regarding the robustness of our method and the tuning of the $(A, B)$ configuration settings. We intend to explore these aspects more thoroughly in our ongoing and future research endeavors, ensuring a comprehensive examination and refinement.
> > >
> > > Thank you again for your constructive feedback, which has been instrumental in enhancing our work.
> > >
> > > Sincerely,
> > >
> > > Authors

---

> ### Author Response · Authors · 2023-11-16
> **Q: Minor suggestion on writing**
>
> **Clarification on Concept Introduction:**
>
> Your observation about the introduction section is well-received. Our intention was to draw parallels between existing PEFT methods (MoLoRA, Poly, MHR) and MoE concepts (MoE, MMoE, CGC, PLE) to provide a quick overview of how our method (CPoly) aligns with and differs from these approaches. We will ensure that each concept is introduced more gradually and is clearly contextualized.
>
> **Clarification on Equation Notation:**
>
> Regarding your point about the notation in Equation 5, we acknowledge the confusion caused by using $A$ and $B$, and have changed it to $W_{down}$ and $W_{up}$ for clarification.

---

> ### Author Response · Authors · 2023-11-18
> **W2.3: From the results in Table 2, it seems that the performance improvement is not consistent**
>
> Thank you for your professional insight. We have also noticed this point. Considering the significant difference in parameter size between FLAN-T5-Large and GLM-10B, we believe that when the parameter size is large, the ability to learn representations is stronger, which can support better fitting of more tasks.
>
> However, the parameter size of FLAN-T5-Large is insufficient to support effective optimization of all tasks. In the Appendix, we have added comparative experiments on different PEFT methods on FLAN-T5-XL on SuperGLUE.
>
> The experimental results have confirmed that after expanding the parameter size, the model's ability to fit multiple tasks becomes stronger, and our method effectively reduces negative transfer phenomena, achieving robust improvements in average and subtasks.
>
> The appendix table is shown as:
> |             | **avg** | **boolq** | **cb** | **copa** | **multirc** | **rte** | **wic** | **wsc.fixed** |
> |-------------|---------|-----------|--------|----------|-------------|---------|---------|---------------|
> | lora        | 82.10   | 89.57     | 78.57  | 96.00    | 79.62       | 87.36   | 67.65   | 75.96         |
> | moe-lora    | 82.97   | 89.32     | 78.57  | 96.00    | 83.76       | 89.16   | 68.02   | 75.96         |
> | poly-lora   | 84.29   | 89.51     | 80.35  | 96.00    | 84.63       | 90.97   | 68.80   | 79.80         |
> | mhr-lora    | 84.39   | 89.51     | 80.35  | 97.00    | 84.61       | 90.97   | 68.49   | 79.80         |
> | cpoly-lora  | **84.94**   | 89.55     | **82.14**  | **97.00**    | **84.96**       | **91.69**   | 68.50   | **80.76**         |

---

### Official Review · Reviewer_WuXZ · 2023-11-02

**Soundness:** 2 fair
**Presentation:** 3 good
**Contribution:** 2 fair
**Rating:** 6
**Confidence:** 3

**Summary:**

The paper proposes a new multi-task learning architecture based on LoRA finetuning framework. They propose the task-common skill sets and task-specific skill sets. Also they learn the task-specific combination weights of task common skill sets using Gumbel-Sigmoid. In the experiment, they adopt two different architectures (encoder-decoder -- T5 and decoder-only -- GLM) and show their performance on several multi-task benchmark in NLP.

**Strengths:**

1. The paper clearly states the difference between their model architecture and previous baseline models.
2. With GLM model, their proposed method outperforms the baseline models for a significant gap.
3. They compare to other baseline models in a fair way.
4. The paper is well-written and easy to follow.

**Weaknesses:**

1. The method does not achieve significant improvement with T5 architecture.  (But I am not expert in NLP tasks and I am open to other reviewers' opinions regarding the performance.)
2. The model design seems incremental to the previous methods Poly and MHR.
3. It is unclear about the optimization loss function and the main paper does not discuss this.

**Questions:**

1. Since you have used Gumbel Sigmoid to optimize the w_i, what is the distribution of all w_i's learnt in the model? Is there any specific loss to force w_i to be close to 0 or 1? What is the performance variance if you need sample the w_i based on Bernouli distributions in the final evaluation?

2. In the experiment results, it is clear that the method with GLM-10B outperforms baselines a lot and the method with T5 stands close to the baselines. In the paper, the authors claim the difference is due to the different model architecture. However, GLM-10B and T5_large also differ a lot in the model capacity. How do you know the difference of performance compared to the baselines is due to the architecture instead of model capacity?

---

> ### Author Response · Authors · 2023-11-16
> **W1/Q2: ‒ The method does not achieve significant improvement with T5 architecture. ‒ How do you know the difference of performance compared to the baselines is due to the architecture instead of model capacity?**
>
> Thank you for your observations regarding the performance improvements with the T5 architecture and the distinction between the performance with GLM-10B and T5 models.
>
> To discern the impact of architecture versus model capacity, we conducted **Ablation Study 3**, which included a range of T5 and FLAN-T5 models with varying capacities (Large/0.78B , XL/3B, XXL/11B) - the XXL version is comparable with GLM-10B. The results demonstrated that our method consistently outperforms other methods across these scales. For instance, with T5-XXL, our method achieved significant gains in performance, suggesting that while model capacity plays a role, the architecture of our method effectively utilizes this additional capacity. On the other hand, the performance saturation observed, especially with FLAN-T5, might partly stem from an overlap between its training dataset and the SuperNI tasks.

---

> ### Author Response · Authors · 2023-11-16
> **W2: The model design seems incremental to the previous methods Poly and MHR.**
>
> Thank you for your comment regarding the comparison of our model design with previous methods like Poly and MHR. While our approach builds upon the foundations laid by these methods, it also introduces significant advancements that distinguish it from its predecessors.
>
> A key innovation in our method is the strategic integration of task-specific skills alongside task-common skills. This dual-skill framework is not just an extension but a significant departure from methods like Poly and MHR, which predominantly focus on shared skills across tasks. Our approach enables a more nuanced and targeted application of skills to specific tasks, enhancing overall performance and adaptability.
>
> Our ablation studies demonstrate the efficacy of this approach. For example, **Ablation Study 1** showed notable improvements in performance with the optimal balance of task-common and task-specific skills, highlighting the effectiveness of our model's unique configuration.
>
> In summary, while our model design is inspired by earlier methods, it offers a more flexible and scalable solution for complex multi-task environments.

---

> ### Author Response · Authors · 2023-11-16
> **W3: It is unclear about the optimization loss function and the main paper does not discuss this.**
>
> Thank you for bringing to our attention the lack of the optimization loss function in our manuscript. We apologize for this oversight. In our method, we have employed the cross-entropy loss function, a standard and widely used choice for language models. The language model predicts next tokens autoregressively. It's essentially a multi classification problem, where the classes are the choices of tokens in the sequence.

---

> ### Author Response · Authors · 2023-11-16
> **Q1: Since you have used Gumbel Sigmoid to optimize the w_i, what is the distribution of all w_i's learnt in the model? Is there any specific loss to force w_i to be close to 0 or 1? What is the performance variance if you need sample the w_i based on Bernouli distributions in the final evaluation?**
>
> During training, we utilize the Gumbel Sigmoid, or Relaxed Bernoulli in binary cases, to smooth the logits derived from w_i. This method enables differentiable sampling from a Bernoulli distribution, crucial for gradient-based optimization in neural networks. In the evaluation phase, we apply a Sigmoid function for smoothing. It's noteworthy that we didn't implement a specific loss function to directly influence this sampling phase. Recognizing the potential impact of such a loss function, we see this as a valuable direction for future research.

---

### Official Review · Reviewer_soi2 · 2023-11-02

**Soundness:** 3 good
**Presentation:** 3 good
**Contribution:** 3 good
**Rating:** 6
**Confidence:** 4

**Summary:**

The paper tackles the task of parameter efficient fine-tuning via Low-Rank Adapter (LoRA) which parametrized the weight updates as low-rank composition to significantly reduce the number of learnable parameters and computation.
The work proposes a novel approach, called Customized Polytropon, which extends LoRA to Multi-Task Learning setup by learning multiple LoRA adapter corresponding to different tasks.
The key insight is to decompose the learnable parameter into Task-Common adapters, which is shared among all tasks, and Task-Specific adapters specifically trained for different tasks.
Given this set of adapters, the proposed framework would learn to combine them for different task, thus enable knowledge transfer among tasks while efficiently learning these adapters.
The paper conducts experiments on SuperCLUE and Super Natural Instruction datasets as well as on T5-Large and GLM-10B models to show its effectiveness.

**Strengths:**

+ The paper is self-contained which makes it accessible to a wide range of reader. Moreover, the paper is also easy to follow.
+ The proposed method of decomposing learnable parameter into a task-common and task-specific portions is sensible as well as easy to develop in practical setting.
+ The problem of parameter efficient finetuning is impactful which helps to democratize LLM technology on consumer device.

**Weaknesses:**

+ As the adapter in LORA are low-rank linear projection of parameters in attention modules, a combination of task-common and task-specific LORA adapters seems to be equivalent to just Mixture of LORA as addition of linear transformations is still a linear transformation (rows 2 and 3). Thus, it is unclear why decomposing learnable parameters would improve performance.

+ The experimental section misses some studies to show the effective of hyper-parameters in the model. For examples, what is the how number of adapters in task-common and task-specific modules effect the performances? What is the impact of rank or the number of tasks toward final performance? These experiments would offer better insight into how robust the proposed method is under different setting.

+ The performance seems to saturate when being applied to strong base model such as T5-Large. It seems to suggest that the effect of task-common components vanish when the base model can generalize well toward different downstream tasks. This could be an interesting phenomena can be study using stronger base model such as llama and llama2.

**Questions:**

+ Could the authors clarify on how they combine different adapters? This would help with understanding as well as reproducibility.
+ Experiments on the effects of the number of adapters/ranks/tasks could provide better insight on the robustness and limitations of the proposed methods
+ Stronger base model can be used to stress-test the generalizability of the proposed framework.

---

> ### Author Response · Authors · 2023-11-16
> **W1: As the adapter in LORA are low-rank linear projection of parameters in attention modules, a combination of task-common and task-specific LORA adapters seems to be equivalent to just Mixture of LORA as addition of linear transformations is still a linear transformation (rows 2 and 3). Thus, it is unclear why decomposing learnable parameters would improve performance.**
>
> Thank you for your insightful observation about the linear nature of these transformations. However, the key advantage of our approach lies in the strategic weighting and targeted application of these adapters.
>
> 1. **Task-Specific Tailoring**: The introduction of task-specific adapters enables our model to customize its responses to the unique requirements of individual tasks, a level of specificity and nuance that general, task-agnostic adapters may not achieve. This allows the model to effectively address task-specific features and variations.
>
> 2. **Dynamic Updating of Components**: The task-common adapters are updated based on the entire multi-task dataset, providing a broad base of learning, while each task-specific adapter is fine-tuned using only data from its respective individual task. This dual approach ensures both generalization and specialization in the model's learning process.
>
> 3. **Evidence from Experiments**: Our experimental results demonstrate that this method of decomposing and dynamically recombining parameters leads to measurable performance improvements.
>
> In conclusion, our model's performance gains stem from the thoughtful combination of task-common and task-specific adapters. This approach not only leverages the strengths of each adapter type but also ensures a more nuanced and effective adaptation to a diverse array of tasks.

---

> ### Author Response · Authors · 2023-11-16
> **Q1: Could the authors clarify on how they combine different adapters? This would help with understanding as well as reproducibility.**
>
> To clarify, we employ linear combinations for combining the output of different adapters in our model, as detailed in equation (1) of our manuscript. To further aid in understanding and reproducibility, we are planning to open-source our code.

---

> ### Author Response · Authors · 2023-11-16
> **W2/Q2: The experimental section misses some studies to show the effective of hyper-parameters in the model; Experiments on the effects of the number of adapters/ranks/tasks could provide better insight on the robustness and limitations of the proposed methods**
>
> Thank you for pointing out the necessity of further experimental insights into the effectiveness of various hyper-parameters in our model. We agree that understanding the impact of the number of adapters, tasks, and rank is crucial for assessing the robustness and limitations of our approach. In response to your concerns, we refer to **Ablation Studies 1 and 2**, which delve into these aspects:
>
> **Ablation Study 1**: This study explored how the number of adapters in task-common and task-specific modules affects performance. For instance, in our experiments with all 3 models, we observed a notable performance increase when using 3 common and 1 task-specific skill compared to 4 common skills. However, further increasing the number of task-specific skills showed diminishing returns.
>
> **Ablation Study 2**: Our method consistently outperformed others across these varying task numbers, demonstrating robust scalability.
>
> Regarding the impact of rank on our model's performance, we based our experimental setup on insights from the [original LoRA paper](https://arxiv.org/abs/2106.09685). In selecting LoRA as our adapter unit, we were guided by the findings in their study, which indicated that variations in rank might not have a significant impact on model performance for each individual LoRA.

---

> ### Author Response · Authors · 2023-11-16
> **W3/Q3:The performance seems to saturate when being applied to strong base model such as T5-Large. It seems to suggest that the effect of task-common components vanish when the base model can generalize well toward different downstream tasks; ● Stronger base model can be used to stress-test the generalizability of the proposed framework.**
>
> As mentioned in our general comment, the base model used in our original manuscript was FLAN-T5-Large. The performance saturation noted may be partly due to the overlap between FLAN-T5's training dataset and the tasks in the SuperNI dataset. To mitigate this and improve comparability, we expanded our ablation studies, particularly **Ablation Study 3**, to encompass a broader range of models including T5-Large (0.78B), T5-XL (3B), and stronger one T5-XXL (11B), and their FLAN-T5 counterparts. These studies demonstrate that as the model scale increases, our method continues to outperform others, indicating its effectiveness across different model capacities.
>
> Regarding the suggestion to use models like Llama and Llama2 for stress-testing our framework, we find it intriguing. Our choice to focus on T5 and GLM was to enable direct comparison with existing results in the manuscript. Exploring our approach with advanced model is an important direction for our future research to further understand and enhance our methodology.

---

### Official Review · Reviewer_yvMZ · 2023-11-03

**Soundness:** 3 good
**Presentation:** 3 good
**Contribution:** 2 fair
**Rating:** 6
**Confidence:** 4

**Summary:**

This paper proposes an approach called Customized Polytropon (C-Poly) for multi-task learning using parameter-efficient modules. The key idea is to explicitly separate task-common skills that can be shared across tasks, and task-specific skills that are unique to each task.  Note that this augments the previously published Poly method (by introducing task-specific adapters, combining subsets of shared and specific tasks, and can allow better interpretation by the parameters learned for selection and weighting).

The model consists of components related to task-common skills (shared low-rank across tasks), and low-rank adapters for each task.  Low-rank adapters (E.g. LoRA) are used to improve param efficienty also.  This approach appears to mitigate negative transfer effects, and improve learning over compared methods.


//Having read the responses: I think its a nice idea, the results are good, but some more analysis / polishing of the paper could be useful before this paper is published (See also response to comments).  To be frank I`m still borderline about this.  I will however increase my score to reflect the authors effort in responses and updates.

**Strengths:**

- this paper presents an intuitive approach to combine shared and specialized skills

- results are good in comparison to previous methods

- sample efficiency is improved; the approach of explicitly separating task-common and task-specific skills to mitigate negative transfer

- offers some more interpretability due to selecting particular skills

**Weaknesses:**

The method introduces additional hyperparameters like number of common/task-specific skills which may require tuning

It is not clearly analyzed if certain tasks benefit more from common or specialized skills

The interpretability via learned task hierarchies is not explored much in experiments

Results are comparable largely to previous work

This paper is an extension of a previous work, with some (nice) but perhaps small increments.

**Questions:**

How is the number of common and task-specific skills determined? Is there a systematic way to set these hyperparameters?

How does performance scale with increasing number of tasks?

How are skills initialized?  does this affect selection?

Can you provide ablation studies controlling for common vs task-specific skills?

More in-depth discussion / results on interpretability

If new tasks appear after training, can the model adapt?

Results are comparable largely to previous work

---

> ### Author Response · Authors · 2023-11-16
> **W1+W2, Q1+Q4: The method introduces additional hyperparameters like number of common/task-specific skills which may require tuning. It is not clearly analyzed if certain tasks benefit more from common or specialized skills. How is the number of common and task-specific skills determined? Is there a systematic way to set these hyperparameters? Can you provide ablation studies controlling for common vs task-specific skills?**
>
> Thank you for highlighting the concerns regarding the number of task-common and task-specific skills. Your query is directly addressed in our **Ablation Study 1**, which provides empirical evidence for the effective allocation of these skills.
>
> Heuristically, we proposed that the total number of skills should be significantly less than the number of tasks to maintain efficiency. For the allocation of two sets of skills, our Ablation Study 1 revealled an optimal balance between common and task-specific skills.
>
> For instance, in the T5-Large model, shifting from 4 common + 0 task-specific skills to a configuration with 3 common and 1 task-specific skill improved the ROUGE-L score from 42.05 to 48.50. However, increasing task-specific skills further (2 common and 2 task-specific) actually resulted in a lower ROUGE-L score of 47.94.

---

> ### Author Response · Authors · 2023-11-16
> **Q2: How does performance scale with increasing number of tasks?**
>
> In response to your question on scalability, **Ablation Study 2** demonstrates that our model robustly scales with an increasing number of tasks. We observed consistent superior performance across different task volumes - from 10 to 100 tasks - irrespective of the base model used. This underscores our method's adaptability and effectiveness in diverse multi-task learning environments, maintaining high performance even as task complexity increases.

---

> ### Author Response · Authors · 2023-11-16
> **Q3: How are skills initialized? does this affect selection?**
>
> Thank you for your question regarding the initialization of skills in our model. The initialization process for our model aligns with the standard [LoRA](https://arxiv.org/abs/2106.09685):
>
> - Matrix A is initialized using the default method for nn.Linear, specifically `nn.init.kaiming_uniform_`.
>
> - Matrix B is initialized to zero using `nn.init.zeros_`.
>
> We recognize that the impact of different initialization strategies on model performance is an important area. However, our core contribution is independent of this, if necessary, we can provide additional information.

---

> ### Author Response · Authors · 2023-11-16
> **Q6: If new tasks appear after training, can the model adapt?**
>
> Thank you for raising an important question about the adaptability of our model to new tasks introduced after training. You're correct in noting that transfer learning for new tasks is a significant and distinct area of study, differing considerably from multi-task learning in its approach and challenges. While your question is indeed relevant, it falls outside the current scope of our discussion, which is focused on multi-task learning. We recognize the importance of this aspect and agree that it merits further exploration in future research.

---

> ### Author Response · Authors · 2023-11-19
> **W3/Q5: The interpretability via learned task hierarchies is not explored much in experiments. More in-depth discussion / results on interpretability**
>
> We appreciate your professional insights on the experimental discussion. For the interpretability aspect of the experiment, we supplemented the router visualization comparison of GLM-10B on NI-10, NI-50, and NI-100 in **Appendix A.5**, and the results showed that the allocation vectors we learned had significant discrimination.
>
> Meanwhile, in **Appendix A.5**, we also added a dendrogram for obtaining hierarchy information based on router allocation weights on NI-100 and compared it with the original poly. The results show that after adding custom skills, the learned allocation vectors can make the differentiation of multiple tasks more reasonable, the number of tasks in each implicit category is more balanced, and more pairwise correlations are learned. Improved the data learning efficiency and generation ability of specific tasks.

---

> > ### Comment · Reviewer_yvMZ · 2023-11-22
> > **thanks to the authors for the responses**
> >
> > thanks to the authors for the responses - I agree on most points, and indeed the results look quite promising and good - although at times they can be minimal
> >
> > I do still think that some more explanations or intuitions in the main paper on the points that were mentioned above could be helpful, e.g. to ensure that overfitting by selecting too-specific tasks etc. is avoided and generalization is not hindered.  ,
> >
> > For the figures added (e.g., 7,8,9) - I m not sure that they are helpful without an explanation (E.g, fig 7 shows basically 100 column matrix which is difficult to decipher without some explanation provided by the authors)
> >
> > thanks again for the detailed responses!

---

> > > ### Author Response · Authors · 2023-11-22
> > > **Thank you for your response!**
> > >
> > > Dear Reviewer,
> > >
> > > Thank you again for your insightful feedback.
> > >
> > > In response to your suggestions, we have added a new sub section, **Section 3.3.4**, to the main paper. This section delves into a deeper insights of $\texttt{C-Poly}$, on the underlying reasons for its effectiveness and strategies for optimizing its performance. We specifically address the concerns you raised regarding the need for more detailed explanations and intuitions. Additionally, more explanations for Figures 7, 8, and 9 are added to clarify their relevance and aid in their interpretation.
> > >
> > > We appreciate your valuable input and hope that these additions will make our paper more comprehensive and clear.
> > >
> > > Sincerely,
> > >
> > > Authors

---

### Author Response · Authors · 2023-11-16
**General Response on All Reviewers' Shared Concerns and Clarifications**

Dear Reviewers,

Thank you for your comments! We are deeply grateful for your invaluable feedback on our manuscript. Your detailed and thoughtful comments have been essential in refining our research. This general comment is intended to collectively address the shared concerns raised by you and provide necessary clarifications.

**1. Clarification on Model Variant Used:**

We appreciate the opportunity to clarify an important aspect of our manuscript regarding the model employed in our experiments. While we did use a T5 model to evaluate our method, it was specifically the [FLAN-T5](https://arxiv.org/abs/2210.11416), not the vanilla T5 as might have been implied. This distinction is crucial for the accuracy and understanding of our research. FLAN-T5s have been fine-tuned on more than 1000 additional tasks covering also more languages. We will ensure this is clearly stated and corrected in the revised manuscript. We apologize for any confusion caused by this oversight and are committed to rectifying it.

**2. Addressing Performance Variation Concerns:**

The concerns raised about the performance variations with the numbers of skills, the allocation of task-common and task-specific skills,  model scales, and task counts. are well-noted. In response, we have carried out additional experiments using:

‒ **T5 (t5.1.1.lm100k)**: [google/t5-large-lm-adapt](https://huggingface.co/google/t5-large-lm-adapt), [google/t5-xl-lm-adapt](https://huggingface.co/google/t5-xl-lm-adapt), [google/t5-xxl-lm-adapt](https://huggingface.co/google/t5-xxl-lm-adapt)

‒ **FLAN-T5**: [google/flan-t5-large](https://huggingface.co/google/flan-t5-large), [google/flan-t5-xl](https://huggingface.co/google/flan-t5-xl), [google/flan-t5-xxl](https://huggingface.co/google/flan-t5-xxl)

‒ **GLM-10B**: [THUDM/glm-10b-chinese](https://huggingface.com/THUDM/glm-10b-chinese)

_**We have included detailed results of these experiments below for your convenience and will reference them in our responses to your specific questions. Additionally, we will incorporate these studies into the revised manuscript and its appendix.**_ Our intention is to offer a more thorough understanding of the model's performance across different scenarios, thereby enhancing the overall clarity and depth of our research.

**3. Experiment Settings Update:**

In the original manuscript, our experiment settings were outlined as follows:

‒ *LoRA*: 1 task-common LoRA of rank 8

‒ *MoE, Poly, MHR*: 4 task-common LoRAs of rank 2

‒ *Our Methods*: 4 task-common LoRAs of rank 2, and for each task, 1 task-specific LoRA of rank 2

Following your suggestions for improved comparability, we revised our experiments during the rebuttal phase. All experiments of our methods were redone using **3 task-common LoRA of rank 2 and 1 task-specific LoRA of rank 2**. This revised setting is consistently applied in all our responses and comments, unless explicitly noted otherwise.

We trust that these clarifications and the additional experimental data adequately address your concerns and shed clearer light on our research approach and findings. We remain dedicated to the continuous improvement of our work and welcome any further suggestions or discussions.

---

> ### Author Response · Authors · 2023-11-16
> **Ablation Study 1: task-common / task-specific skill allocation**
>
> Your observation about the number of common and task-specific skills is crucial. To determine the optimal balance between these two types of skills, we have conducted a series of ablation studies. The table below illustrates the results of our ablation studies (with 100 tasks from Super NI) using different configurations of common and task-specific skills across various base models:
> | Base Model    | n_common_skills | n_specific_skills | rouge1 | rougeL | rougeLsum |
> |---------------|-----------------|-------------------|--------|--------|-----------|
> | T5-Large      | 4               | 0                 | 43.04  | 42.05  | 42.09     |
> | T5-Large      | 3               | 1                 | **49.34**  | **48.50**  | **48.51**     |
> | T5-Large      | 2               | 2                 | 48.79  | 47.94  | 47.94     |
> | T5-Large      | 1               | 3                 | 47.46  | 46.60  | 46.59     |
> | FLAN-T5-Large | 4               | 0                 | 68.45  | 67.60  | 67.58     |
> | FLAN-T5-Large | 3               | 1                 | **68.69**  | **67.80**  | **67.82**     |
> | FLAN-T5-Large | 2               | 2                 | 68.47  | 67.63  | 67.65     |
> | FLAN-T5-Large | 1               | 3                 | 68.21  | 67.36  | 67.41     |
> | GLM-10B       | 4               | 0                 | 47.96  | 46.80  | 46.80     |
> | GLM-10B       | 3               | 1                 | **49.53**  | **48.45**  | **48.45**     |
> | GLM-10B       | 2               | 2                 | 49.19  | 48.16  | 48.16     |
> | GLM-10B       | 1               | 3                 | 46.85  | 45.74  | 45.73     |
>
> The results clearly demonstrate that the introduction of task-specific skills significantly enhances model performance. In all models, we observed a notable improvement in Rouge scores when moving from a configuration with only common skills (4 common, 0 task-specific) to a mixed setup (3 common, 1 task-specific).
>
> However, it is important to highlight that the benefits of adding more task-specific skills diminish marginally when we keep the total number of trainable parameters constant. As we have more task-specific skills, the performance gains plateau or even slightly decrease.
>
> In conclusion, our C-Poly approach innovatively integrates task-specific skills with common skills. This dual-skill framework is a significant departure from previous methods that primarily focus on shared skills across tasks. The introduction of task-specific skills is not just an incremental addition but a fundamental shift that allows for more nuanced and effective multi-task learning models. Determining the ideal ratio and configuration of these skills for different types of tasks and datasets is a fascinating area that warrants deeper exploration.

---

> ### Author Response · Authors · 2023-11-16
> **Ablation Study 2: the number of tasks**
>
> To understand how our approach scales with the number of tasks, we performed experiments with varying task numbers - 10, 50, and 100 from SuperNI dataset. The following tables summarizes the results of these experiments:
>
> **FLAN-T5-Large:**
>
> |  Tasks Numbers |  PEFT Method   | rouge1    | rougeL    | rougeLsum |
> |----------------|----------------|-----------|-----------|-----------|
> | 10             | LoRA           | 67.82     | 67.01     | 67.03     |
> | 10             | MoE-LoRA       | 67.95     | 67.12     | 67.15     |
> | 10             | Poly           | 68.10     | 67.29     | 67.33     |
> | 10             | MHR            | 77.49     | 77.25     | 77.36     |
> | 10             | **Our Method** | **77.73** | **77.58** | **77.61** |
> | 50             | LoRA           | 70.66     | 69.10     | 69.03     |
> | 50             | MoE-LoRA       | 70.81     | 69.25     | 69.21     |
> | 50             | Poly           | 70.76     | 69.23     | 69.15     |
> | 50             | MHR            | 70.92     | 69.39     | 69.33     |
> | 50             | **Our Method** | **71.17** | **69.68** | **69.62** |
> | 100            | LoRA           | 68.26     | 67.42     | 67.42     |
> | 100            | MoE-LoRA       | 68.59     | 67.76     | 67.75     |
> | 100            | Poly           | 68.45     | 67.60     | 67.58     |
> | 100            | MHR            | 68.84     | 67.77     | 67.78     |
> | 100            | **Our Method** | **68.69** | **67.80** | **67.82** |
>
> **T5-Large:**
>
> |  Tasks Numbers |  PEFT Method   | rouge1    | rougeL    | rougeLsum |
> |----------------|----------------|-----------|-----------|-----------|
> | 10             | LoRA           | 14.22     | 14.12     | 14.19     |
> | 10             | MoE-LoRA       | 16.75     | 16.71     | 16.76     |
> | 10             | Poly           | 17.34     | 17.31     | 17.38     |
> | 10             | MHR            | 17.17     | 17.11     | 17.16     |
> | 10             | **Our Method** | **42.62** | **42.47** | **42.60** |
> | 50             | LoRA           | 32.58     | 31.64     | 31.59     |
> | 50             | MoE-LoRA       | 39.50     | 38.47     | 38.46     |
> | 50             | Poly           | 46.13     | 44.25     | 44.28     |
> | 50             | MHR            | 47.27     | 45.43     | 45.39     |
> | 50             | **Our Method** | **53.39** | **51.68** | **51.63** |
> | 100            | LoRA           | 34.16     | 33.64     | 33.65     |
> | 100            | MoE-LoRA       | 36.82     | 36.13     | 36.15     |
> | 100            | Poly           | 43.04     | 42.05     | 42.09     |
> | 100            | MHR            | 44.24     | 43.32     | 43.34     |
> | 100            | **Our Method** | **49.34** | **48.50** | **48.51** |
>
> **GLM-10B:**
>
> |  Tasks Numbers |  PEFT Method   | rouge1    | rougeL    | rougeLsum |
> |----------------|----------------|-----------|-----------|-----------|
> | 10             | LoRA           | 30.64     | 30.40     | 30.42     |
> | 10             | MoE-LoRA       | 33.92     | 33.79     | 33.77     |
> | 10             | Poly           | 34.53     | 34.41     | 34.31     |
> | 10             | MHR            | 33.63     | 33.47     | 33.47     |
> | 10             | **Our Method** | **43.74** | **43.72** | **43.65** |
> | 50             | LoRA           | 34.16     | 33.00     | 32.98     |
> | 50             | MoE-LoRA       | 39.87     | 38.63     | 38.55     |
> | 50             | Poly           | 44.81     | 43.09     | 43.07     |
> | 50             | MHR            | 45.32     | 43.62     | 43.56     |
> | 50             | **Our Method** | **53.17** | **51.27** | **51.32** |
> | 100            | LoRA           | 43.16     | 42.04     | 42.09     |
> | 100            | MoE-LoRA       | 45.97     | 44.79     | 44.89     |
> | 100            | Poly           | 47.96     | 46.80     | 46.80     |
> | 100            | MHR            | 48.53     | 47.34     | 47.33     |
> | 100            | **Our Method** | **49.53** | **48.45** | **48.45** |
>
> These results affirm that our model is highly effective and scalable across various task numbers. Our approach shows not only superior performance in lower task settings but also maintains its competitive edge as the number of tasks increases.

---

> ### Author Response · Authors · 2023-11-16
> **Ablation Study 3: model scale**
>
> Thank you for your insightful question concerning the comparative performance of our method using GLM-10B and T5 models, and whether these differences are attributable to model architecture or model capacity. To address this query, we conducted an ablation study across various scales of the T5 model, including T5-Large (0.78B), T5-XL (3B), and T5-XXL (11B), as well as FLAN-T5 variants. The following experiments were conducted using 100 tasks from SuperNI.
>
> **T5:**
>
> | Base Model | PEFT Method | rouge1 | rougeL | rougeLsum |
> |------------|-------------|--------|--------|-----------|
> | T5-Large   | LoRA        | 34.16  | 33.64  | 33.65     |
> | T5-Large   | MoE-LoRA    | 36.82  | 36.13  | 36.15     |
> | T5-Large   | Poly        | 43.04  | 42.05  | 42.09     |
> | T5-Large   | MHR         | 44.24  | 43.32  | 43.34     |
> | T5-Large   | **Our Method**  | **49.34**  | **48.50**  | **48.51**     |
> | T5-XL      | LoRA        | 34.93  | 34.34  | 34.40     |
> | T5-XL      | MoE-LoRA    | 39.78  | 38.83  | 38.87     |
> | T5-XL      | Poly        | 43.61  | 42.61  | 42.62     |
> | T5-XL      | MHR         | 45.53  | 44.62  | 44.61     |
> | T5-XL      | **Our Method**  | **50.57**  | **49.74**  | **49.76**     |
> | T5-XXL     | LoRA        | 49.97  | 48.89  | 48.93     |
> | T5-XXL     | MoE-LoRA    | 52.14  | 51.12  | 51.15     |
> | T5-XXL     | Poly        | 55.42  | 54.65  | 54.64     |
> | T5-XXL     | MHR         | 55.81  | 55.01  | 55.01     |
> | T5-XXL     | **Our Method**  | **62.23**  | **61.44** | **61.47**     |
>
> **FLAN-T5:**
>
> | Base Model    | PEFT Method | rouge1 | rougeL | rougeLsum |
> |---------------|-------------|--------|--------|-----------|
> | FLAN-T5-Large | LoRA        | 68.26  | 67.42  | 67.42     |
> | FLAN-T5-Large | MoE-LoRA    | 68.59  | 67.76  | 67.75     |
> | FLAN-T5-Large | Poly        | 68.45  | 67.60  | 67.58     |
> | FLAN-T5-Large | MHR         | 68.84  | 67.77  | 67.78     |
> | FLAN-T5-Large | **Our Method**  | **68.69**  | **67.80**  | **67.82**     |
> | FLAN-T5-XL    | LoRA        | 71.01  | 70.21  | 70.24     |
> | FLAN-T5-XL    | MoE-LoRA    | 71.08  | 70.29  | 70.33     |
> | FLAN-T5-XL    | Poly        | 71.12  | 70.31  | 70.35     |
> | FLAN-T5-XL    | MHR         | 71.18  | 70.36  | 70.40     |
> | FLAN-T5-XL    | **Our Method**  | **71.57**  | **70.72**  | **70.74**     |
> | FLAN-T5-XXL   | LoRA        | 71.89  | 71.07  | 71.08     |
> | FLAN-T5-XXL   | MoE-LoRA    | 72.05  | 71.25  | 71.26     |
> | FLAN-T5-XXL   | Poly        | 72.55  | 71.76  | 71.78     |
> | FLAN-T5-XXL   | MHR         | 72.40  | 71.61  | 71.63     |
> | FLAN-T5-XXL   | **Our Method**  | **73.09**  | **72.27**  | **72.28**     |
>
> - The results demonstrate a clear trend: as the model scale increases, there is a consistent improvement in performance for all methods, including ours.
> - Notably, our method shows significant gains over other PEFT methods across all scales of the T5 and FLAN-T5 models, suggesting that our approach effectively leverages the increased model capacity.
> - The performance improvement is more pronounced in larger models (T5-XXL and FLAN-T5-XXL), indicating that while model capacity plays a role, our method might be particularly effective at utilizing the additional capacity.
> - These findings suggest that the performance improvement of our method is not solely due to the increased model capacity.

---

### Author Response · Authors · 2023-11-21
**Thanks again and looking forward to your feedback!**

Dear Reviewers,

Thank you once again for your constructive comments on our paper. Your insights have been instrumental in enhancing the quality of our work. We recognize the demanding nature of the review process and deeply appreciate the time and expertise you invest in this crucial task.

In response to your feedback, we have thoroughly addressed all the vital points raised. We are eager to hear your thoughts, as your continued feedback is crucial to ensure our submission aligns with the conference's high standards and allows for any necessary timely revisions.

We remain committed to a productive dialogue and are open to any further discussion or clarification that might be needed to refine our research further. Thanks again!

Warm regards,

Authors

---

### Public Comment · ~Raman_Dutt1 · 2024-08-12
**Accompanied Code for the Paper**

I would like to thank the authors for this insightful and interesting work. To facilitate the usability of this work, could the authors provide the code for the same.

---

### Meta-Review · Area_Chair_fDdi · 2023-12-09

**Metareview:**

The meta-reviewer has carefully read the paper, reviews, rebuttals, and discussions between authors and reviewers. The meta-reviewer agrees with the reviewers that this is a good submission to ICLR. The paper introduces Customized Polytropon (C-Poly), an approach for multi-task learning that enhances the Poly method by integrating task-specific adapters to differentiate between familiar and unique task skills. C-Poly uses low-rank adapters, such as LoRA, for each task to ensure parameter efficiency and to counter adverse transfer effects. The model combines shared knowledge across tasks with task-specific knowledge through a soft combination of shared and specific LoRA modules, with binary combination weights indicating expert activation. C-Poly is tested on T5-Large and GLM-10B models using SuperGlue and Super Natural Instructions datasets, showing improvements over recent methods like LoRa, MoE-LoRA, and MHR. The meta-reviewer agrees with the reviewers that the paper can interest the multitask learning community and recommends acceptance.

**Justification For Why Not Higher Score:**

N/A

**Justification For Why Not Lower Score:**

N/A

---

### Decision · Program_Chairs · 2024-01-16

Accept (poster)